# Variational Automatic Curriculum Learning for Sparse-Reward Cooperative Multi-Agent Problems

**Jiayu Chen** [1♯]**, Yuanxin Zhang**[1]**, Yuanfan Xu**[1]**, Huimin Ma**[3]**,**
**Huazhong Yang**[1]**, Jiaming Song**[4]**, Yu Wang**[1]**, Yi Wu**[12♮]
[1] Tsinghua University, [2] Shanghai Qi Zhi Institute,
[3] University of Science and Technology Beijing, [4] Stanford University,
[♯]jia768167535@gmail.com, [♮]jxwuyi@gmail.com

## Abstract

We introduce a curriculum learning algorithm, *Variational Automatic Curriculum Learning* (VACL), for solving challenging goal-conditioned cooperative multi-agent reinforcement learning problems. We motivate our paradigm through a variational perspective, where the learning objective can be decomposed into two terms: task learning on the current task distribution, and curriculum update to a new task distribution. Local optimization over the second term suggests that the curriculum should gradually expand the training tasks from easy to hard. Our VACL algorithm implements this variational paradigm with two practical components, task expansion and entity progression, which produces training curricula over both the task configurations as well as the number of entities in the task. Experiment results show that VACL solves a collection of sparse-reward problems with a large number of agents. Particularly, using a single desktop machine, VACL achieves 98% coverage rate with 100 agents in the simple-spread benchmark and reproduces the ramp-use behavior originally shown in OpenAI's hide-and-seek project. Our project website is at `https://sites.google.com/view/vacl-neurips-2021`.

## 1 Introduction

Building intelligent agents in complex multi-agent games remains a long-standing challenge in artificial intelligence [23]. Recently, it has been a trend to apply multi-agent reinforcement learning (MARL) to extremely challenging multi-agent games, such as Dota II [5], StarCraft II [29] and Hanabi [3]. Despite these successes, learning intelligent multi-agent policies in general still remains a great RL challenge. Multi-agent games allow sophisticated interactions between agents and environment. Feasible solutions may require non-trivial intra-agent coordination, which leads to substantially more complex strategies than the single-agent setting. Moreover, as the number of agents increases, the joint action space grows at an exponential rate, which results in an exponentially large policy search space. Thus, most existing MARL applications typically require shaped rewards, or assume simplified environment dynamics, or only handle a limit number of agents.

We tackle goal-conditioned cooperative MARL problems with *sparse* rewards through a novel variational inference perspective. Assuming each task can be parameterized by a continuous representation, we introduce a variational proposal distribution over the task space and then decomposing the overall training objective into two separate terms, i.e., a policy improvement term and a task proposal update term. By treating the proposal distribution updates as the training curriculum, such a variational objective naturally suggests a curriculum learning framework by alternating between curriculum update and MARL training. In addition, we propose a continuous relaxation technique for discrete variables so that the variational curricula can be also applied over the number of agents and objects in the task, which leads to a generic and unified training paradigm for the cooperative MARL setting.

35th Conference on Neural Information Processing Systems (NeurIPS 2021).

We implement our variational training paradigm through a computationally efficient algorithm, *Variational Automatic Curriculum Learning (VACL)*. VACL consists two components, *task expansion* and *entity progression*, which generate a series of effective training tasks in a hierarchical manner. Intuitively, *entity progression* leverages the inductive bias that tasks with more entities are usually more difficult in MARL and, therefore, progressively increases the entity size in the environment. *Task expansion* assumes a fixed number of entities and is motivated by Stein variational gradient descent (SVGD) [17] to efficiently expand the task distribution towards the entire task space.

We provide thorough discussions on the differences between VACL and existing works (Sec. 3.4) and empirically validate VACL using a single desktop machine in four physical multi-agent cooperative games, i.e., two in the multi-agent particle-world environment (MPE) [19] and two in the MuJoCo-based hide-and-seek environment (HnS) [2] (Sec. 4). In the MPE games, VACL learns significantly faster and better than all baselines and achieves over **98% coverage rate** with **100 agents** in the *simple-spread* testbed even using **sparse rewards**. In the HnS scenarios, VACL achieves over 90% success rates on both two games, including reproducing the *ramp use* behavior discovered in [2] and solving a sparse-reward multi-agent *Lock-and-Return* challenge where none of the baselines can produce a success rate above 15% using the same amount of training timesteps.

## 2 A Variational Perspective on Curriculum Learning

### 2.1 Preliminary

We formulate each goal-conditioned cooperative multi-agent task as a multi-agent Markov decision process (MDP), $M(n, \phi) = <n, \phi, \mathcal{S}, \mathcal{A}, \mathcal{O}, O, P, R, s_\phi^0, g_\phi, \gamma>$, parameterized by a discrete variable $n$ (e.g., the number of agents[1]) and a continuous parameter $\phi \in \Phi$ (i.e., the initial state and goal; details in App. C.2). $\mathcal{S}$ is the state space. $\mathcal{A}$ is the shared action space for each agent. $\mathcal{O}$ is the observation space. $o_i = O(s; i)$ denotes the observation for agent $i$ under state $S$. $s_\phi^0$ denotes the initial state and $g_\phi$ denotes the goal. $P(s'|s, A)$ and $R(s, A; g_\phi)$ are the transition probability and the goal-conditioned reward function given state $s$ and joint actions $A = (a_1, \ldots, a_n)$. $\gamma$ is the discounted factor.

We consider homogeneous agents and learn a shared goal-conditioned policy $\pi_\theta(a_i|o_i, g)$ parameterized by $\theta$ for each agent $i$. The final objective is to maximize the expected accumulative reward over the entire task space, i.e.,

$$J(\theta) = \mathbb{E}_{n, \phi, a_i^t, s^t} \left[ \sum_t \gamma^t R(s^t, A^t; g_\phi) \right] = \mathbb{E}_{n, \phi}[V(n, \phi, \pi_\theta)], \tag{1}$$

where $V(n, \phi, \pi_\theta)$ denotes the value function for $\pi_\theta$ over task $M(n, \phi)$. We remark that for problems with 0/1 sparse rewards, $V(n, \phi, \pi_\theta)$ can be also interpreted as the success rate of $\pi_\theta$[2].

The main idea of curriculum learning is to construct a task sampler $q(n, \phi)$ that always generates training tasks $M(n, \phi)$ that yield the largest learning progress for $\pi_\theta$ to efficiently maximize $J(\theta)$.

### 2.2 Stein Variational Inference over a Continuous Task Space

For simplicity, we ignore the discrete variable $n$ and only focus on the continuous variable $\phi$ in this subsection. Herein, our objective is to maximize the expectation $\mathcal{L} = \mathbb{E}_{\phi \sim p(\phi)} [V(\phi, \pi)]$, where $p(\phi)$ is a uniform distribution over feasible values of $\phi$. The subscript $\theta$ of $\pi$ is omitted for conciseness.

Given the objective $\mathcal{L}$, we motivate our automated curriculum learning procedure through the lens of variational inference. Let $q(\phi)$ denote the current distribution of training tasks, which will vary throughout learning; we can represent curriculum learning through an objective that optimizes $q(\phi)$.

---

[1]$n$ can also be a discrete vector where each dimension represents the number of a particular type of objects. We now simply assume $n$ is a single integer denoting the agent number for conciseness.

[2]This only rigorously holds when $\gamma = 1$. But let's assume it for simplicity. In practice, this statement approximately holds with a close-to-1 $\gamma$ and a relatively short horizon.

The following lower bound for the original objective $\mathcal{L}$ can be derived:

$$\mathcal{L} = \mathbb{E}_{\phi \sim p}[V(\phi, \pi)] = \mathbb{E}_{\phi \sim q}\left[\frac{p(\phi)}{q(\phi)}V(\phi, \pi)\right] = \mathbb{E}_{\phi \sim q}\left[V(\phi, \pi) + \left(\frac{p(\phi)}{q(\phi)} - 1\right)V(\phi, \pi)\right] \quad (2)$$

$$\geq \underbrace{\mathbb{E}_{\phi \sim q(\phi)}[V(\phi, \pi)]}_{\mathcal{L}_1 : \text{policy update}} + \underbrace{\mathbb{E}_{\phi \sim q(\phi)}\left[V(\phi, \pi) \log \frac{p(\phi)}{q(\phi)}\right]}_{\mathcal{L}_2 : \text{curriculum update}} \quad (3)$$

where the inequality is due to $x - 1 \geq \log x$, with equality achieved at $p(\phi) = q(\phi)$ for all $\phi$.

Now, we have decomposed $\mathcal{L}(\pi, q)$ into a lower bound that contains two terms $\mathcal{L}_1$ and $\mathcal{L}_2$. $\mathcal{L}_1$ is simply a policy update objective under the current task curriculum $q(\phi)$ while the second term $\mathcal{L}_2$ can be interpreted as a curriculum update objective by maximizing $q(\phi)$ w.r.t. the policy $\pi$. Therefore, the decomposed lower bound naturally yields a curriculum learning framework by performing an iterative learning procedure, i.e., alternating between (1) maximizing $\mathcal{L}_1$ with $\pi$ (policy update) and (2) maximizing $\mathcal{L}_2$ with $q(\phi)$ (curriculum update). We remark that we only optimize $\pi$ w.r.t. $\mathcal{L}_1$ instead of $\mathcal{L}_1 + \mathcal{L}_2$ for the purpose of stabilizing policy learning since $q$ can be very different from $p$ in the early stage of training. This also aligns with the curriculum learning paradigm. In the following proposition, we show that if we can perfectly optimize the RL procedure for $\mathcal{L}_1$ under the curriculum $q(\phi)$, then the curriculum objective $\mathcal{L}_2$ will encourage $q(\phi)$ to converge to $p(\phi)$.

**Proposition 1.** *Assume that $V(\phi, \pi) \in [0, 1]$ and that there exists a unique policy $\pi^\star$ such that $V(\phi, \pi^\star) = 1$ for all $\phi$. For any $q \in \mathcal{P}(\Phi) \setminus \{p\}$ (i.e., any curriculum that is not the final one) and $\pi \in \arg\max_\pi \mathcal{L}_1(\pi, q)$ (i.e., a policy that is optimal under the current curriculum), $\mathcal{L}_2(\pi, q) < \mathcal{L}_2(\pi^\star, p) = 0$.*

*Proof.* First, we have that $\mathcal{L}_1(\pi^\star, p) = 1$ (definition of $\pi^\star$) and $\mathcal{L}_2(\pi^\star, p) = 0$. Suppose that $\exists \pi, q$ such that $q \neq p$ and $\mathcal{L}_1(\pi, q) = 1$; then it must be the case that whenever $V(\phi, \pi) \neq 1$, $q(\phi) = 0$, i.e., $\forall \phi \in \text{supp}(q)$, $V(\phi, \pi) = 1$. Then $\mathcal{L}_2(\phi, \pi) = -D_{\text{KL}}(q \| p) < 0$, which completes the proof. $\square$

In other words, if our policy $\pi$ already performs well under the current curriculum $q(\phi)$, then $\mathcal{L}_2$ will encourage $q(\phi)$ to become "closer" to $p(\phi)$, aligning with our eventual target of learning all tasks.

### 2.2.1 Stein Variational Gradient Descent for Curriculum Update

The most straightforward approach for optimizing $\mathcal{L}_2$ is to learn a generative model w.r.t. the current $\pi$, which can be particularly sample-inefficient. Instead, we consider updates to our current curriculum via smooth updates, which is much more efficient than learning generative models.

Let us represent the task distribution under the current curriculum $q(\phi)$ with a set of particles $\mathcal{Q} = \{\phi\}$. To update each particle $\phi$ in $\mathcal{Q}$, we consider an incremental transform $T(\phi) = \phi + \epsilon f(\phi)$, where $f(\phi)$ is a smooth function and $\epsilon$ is a magnitude scalar. Our goal is then to derive suitable updates to $\mathcal{Q}$ that maximizes $\mathcal{L}_2(q, \pi)$ based on the transform $T$. Following [9] and [27], we introduce an assumption over the generalization ability of the task space.

**Assumption 1.** *An optimal policy $\pi$ that is trained over a specific task $\phi$ has some generalization ability to another goal $\phi'$ close to $\phi$. Specifically, if $V(\phi, \pi) = c$ for any $\phi \in \text{supp}(q)$, then $V(\phi', \pi) = c$ for all $\phi' \in \mathcal{B}(\phi)$, where $\mathcal{B}(\phi)$ is a small open ball around $\phi$.*

In the context of curriculum learning, this assumption suggests that for any policy that is trained on a set of tasks, the performance (e.g., success rate) does not change if we only change the task parameters by an infinitesimal amount. In the following theorem, we derive the functional gradient of the curriculum update objective $\mathcal{L}_2$ in a reproducing kernel Hilbert space (RKHS).

**Theorem 1.** *Let $T(\phi) = \phi + \epsilon f(\phi)$ where $f$ is an element of some vector-valued RKHS of a positive definite kernel $k(\phi, \phi') : \Phi \times \Phi \to \mathbb{R}$, and $q_{[T]}$ the density of $\psi = T(\phi)$ when $\phi \sim q$, then*

$$\nabla_f \mathcal{L}_2(q_{[T]}, \pi)|_{f=0} = f^*(\phi), \quad (4)$$

*where $f^*(\cdot) = \mathbb{E}_{\phi' \in \mathcal{Q}}[V(\phi', \pi)(k(\phi', \cdot)\nabla_{\phi'} \log p(\phi') + \nabla_{\phi'} k(\phi', \cdot))]$.*

Proof is in App. A. While the above statement is very similar to that used for Stein variational gradient descent [17], the update for our case also directly depends on the value $V(\phi', \pi)$, and thus indirectly

depends on the current policy. Intuitively, if $V(\phi', \pi)$ is large, then $\phi'$ will contribute more to the curriculum update, whereas if $V(\phi', \pi) = 0$, then $\phi'$ will not contribute to the update.

Since we assume that $p(\phi)$ is uniform over the task space, $\nabla_{\phi'} \log p(\phi') = 0$ for the interior of $\Phi$, and the task update becomes

$$f^*(\cdot) = \mathbb{E}_{\phi' \in \mathcal{Q}}[V(\phi', \pi) \cdot \nabla_{\phi'} k(\phi', \cdot)]. \tag{5}$$

For example, we take the RBF kernel $k(\phi, \phi') = \exp(-\frac{1}{h}\|\phi - \phi'\|_2^2)$, and then Eq. (5) will drive $\phi$ away from neighbouring points that have large $k(\phi, \phi')$, and the repelling force scales with $V(\phi', \pi)$. If $V(\phi', \pi) = 0$, *i.e.*, $\phi'$ has not been solved at all, then there is no repelling force from $\phi'$ to $\phi$. Intuitively, this process encourages the curriculum update to prioritize tasks that have not been fully solved yet, by exploring in the vicinity of existing tasks with high values.

### 2.3 Continuous Relaxation for Discrete Parameter

Now we switch to the discrete parameter $n$ in our task representation, which results in a non-continuous task space and can be difficult to directly combine with continuous optimization techniques. Therefore, we propose to utilize a continuous variable alternative $z$ to represent the discrete variable $n$ so that the same variational inference technique for $\phi$ can be applied here. Specifically, assuming $n$ denotes the number of agents, we represent $n$ with a categorical distribution $p(n; z) = \text{Cat}(z_1, z_2, \ldots, z_N)$ parametrized by a continuous vector $z = [z_1, \ldots, z_N]$ assuming the maximum possible of agents[3] is $N$. $p(n; z)$ denotes the distribution which generates $n$ agents with probability $z_n$. In this case, we can similarly introduce a target distribution $p(z)$ over the space of $z$ and derive the corresponding variational updates with $V(z, \pi)$. Since we typically wish to learn a policy that can generalizes across different number of agents, such a representation over multiple $n$'s naturally satisfies this practical requirement. We also remark that this formulation can be directly extended to the problems with multiple discrete variables (e.g., number of both agents and objects) via a categorical distribution over all possible discrete combinations.

## 3 Variational Automatic Curriculum Learning

In this section, we implement the variational inference paradigm with computation-efficient approximations as our VACL algorithm in the context of sparse-reward cooperative MARL problems.

We will first present *Task Expansion* in Sec. 3.1, which is the most important algorithmic component assuming a continuous task space. Then, *Entity Progression*, i.e., the component specialized for discrete task parameters, will be introduced in Sec. 3.2. Finally, the full algorithm and the literature review will follow in Sec. 3.3 and Sec. 3.4.

### 3.1 Task Expansion as Approximated Stein Variational Inference

*Task Expansion* is a computation-efficient curriculum update method, which assumes a continuous task representation $\phi$ and optimizes the variational lower bound $\mathcal{L}_2$ w.r.t. $q(\phi)$. Directly optimizing $\mathcal{L}_2$ can be computational challenging. In the following content, we will present a few critical techniques for the efficient approximation computation.

### 3.1.1 An Ever-Expanding Particle Set

In the standard Stein variational inference framework, we typically maintain a fixed-size particle set $\mathcal{Q}$ and gradually update the distribution of these particles using Stein variational gradient descent. Since the policy is constantly changing throughout training, we need to repeatedly update the entire set of particles to match the corresponding proposal distribution under $\mathcal{L}_2$, which is computationally expensive. Note that in the cooperative MARL setting, assuming a fixed amount of agents and an ever-improving policy, if we initialize $q(\phi)$ to the proper sub-space of easiest tasks, $q(\phi)$ would monotonically expand towards the entire task space. Hence, we can simply maintain an ever-expanding particle set by repeatedly adding novel tasks with sufficiently high values (i.e., success rates) under the current policy without the need of modifying those existing particles in $\mathcal{Q}$.

---

[3]If $N$ is unbounded, we may consider parametrized discrete random processes such as the Chinese restaurant process. For all practical purposes here, this is unnecessary.

### 3.1.2 Value Quantization

Another issue for optimizing $\mathcal{L}_2$ is that whenever $\pi$ updates, $V(\phi, \pi)$ changes accordingly but evaluating the value function can be particularly expensive in MARL problems. As a practical approximation, we roughly categorize all the tasks into 3 categories w.r.t. their values, i.e., a solved subspace of highest values, $\mathcal{Q}_{\mathrm{sol}} = \{\phi | V(\phi, \pi) > \sigma_{\max}\}$, an active subspace of moderate values, $\mathcal{Q}_{\mathrm{act}} = \{\phi | \sigma_{\min} \leq V(\phi, \pi) \leq \sigma_{\max}\}$, and the remaining unsolved subspace. Here $\sigma_{\min}$ and $\sigma_{\max}$ are quantization thresholds. Since the value function would monotonically increase as training proceeds, we only need to verify whether an active task becomes solved or whether an unsolved task becomes active, and maintain $\mathcal{Q}_{\mathrm{act}}$ and $\mathcal{Q}_{\mathrm{sol}}$ accordingly. Once a task becomes solved,

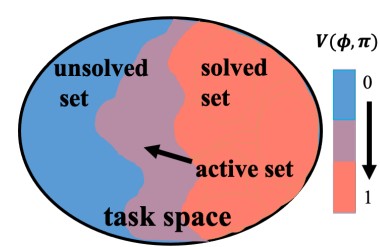

Figure 1: Task space partition.

future value estimation becomes no longer necessary. In addition, w.r.t. the curriculum learning principle, when optimizing $\pi$ under $\mathcal{L}_1$, we can also sample more training tasks from $\mathcal{Q}_{\mathrm{act}}$ for more effective training.

### 3.1.3 Sampling-Based Particle Exploration

With an ever-expanding particle set $\mathcal{Q}$, we need to explore novel task samples in the task space and add those tasks with sufficiently high values (i.e., $V \geq \sigma_{\min}$) to $\mathcal{Q}$. The update equation for $q(\phi)$ in Eq.(5) naturally suggests a diversity-based task expansion scheme, namely exploring sufficiently novel tasks in the vicinity of $\mathcal{Q}$. Due to the difficulty in value estimation, we again follow the value quantization framework and simplify Eq.(5) as follows:

$$\tilde{f}^*(\cdot) \propto \mathbb{E}_{\phi' \in \mathcal{Q}_{\mathrm{sol}}}[\nabla_{\phi'} k(\phi', \cdot)]. \quad (6)$$

In the simplified Eq.(6), we simply ignore tasks with moderate values (i.e., active tasks in $\mathcal{Q}_{\mathrm{act}}$) and only evaluate the expectation w.r.t. tasks with the highest values (i.e., solved tasks in $\mathcal{Q}_{\mathrm{sol}}$) as a practical approximation. Moreover, since $\mathcal{Q}_{\mathrm{sol}}$ is ever-expanding, Eq.(6) also suggests that we should explore novel tasks in the

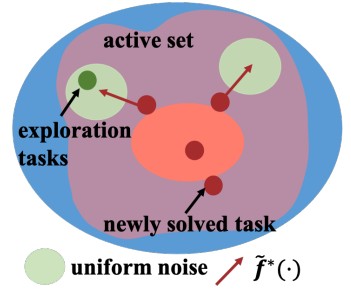

Figure 2: We explore novel tasks in the boundary region between $\mathcal{Q}_{\mathrm{sol}}$ and $\mathcal{Q}_{\mathrm{act}}$

boundary region between $\mathcal{Q}_{\mathrm{sol}}$ and $\mathcal{Q}_{\mathrm{act}}$ (Fig. 2). Hence, in practice, whenever an active task from $\mathcal{Q}_{\mathrm{act}}$ becomes solved after policy update, we select them as seed tasks to derive novel exploration task samples.

Eq.(6) suggests purely gradient-based exploration, which may cause practical issues in curriculum learning since the feasible task parameterization space is often highly constrained. Fig. 5(b) illustrates a maze scenario where agents need to navigate towards the landmarks and avoid the black obstacles. The task parameterization $\phi$ includes the positions of agents and landmarks. The feasible subspace of $\phi$ is constrained by the obstacles while these feasibility constraints are only accessible as a black box in the curriculum learning setting. Therefore, directly computing a constraint-free gradient can easily lead to an exploration direction towards infeasible regions. Here we suggest a simple sampling-based enhancement for such highly-constrained problems, which promotes effective task-space exploration and can be viewed as a zero-th order approximation of the projected gradient.

**Rejection-Sampling Exploration:** We explore novel tasks $\phi_{\mathrm{exp}}$ by adding small uniform noise to the gradient of seed tasks, namely $\phi_{\mathrm{exp}} \leftarrow \phi_{\mathrm{seed}} + \epsilon \tilde{f}^*(\phi_{\mathrm{seed}}) + \mathrm{Unif}(-\delta, \delta)$. Then we reject those infeasible tasks by querying the environment for configuration feasibility.

**Algorithm 1:** The VACL Algorithm

---

**Input:** $\theta, n_0, N, B, B_{\exp}$; // $B$: training batch size; $B_{\exp}$: exploration size
**Output:** final policy $\pi_\theta$;
$k \leftarrow 0, \mathcal{Q}_{\text{sol}}^0 \leftarrow \{\}, \mathcal{Q}_{\text{act}}^0 \leftarrow$ **GetEasy**$(n_0)$;  // use easy tasks to initialize $q(\phi)$
**repeat**

    $z \leftarrow 1, n_{k+1} \leftarrow$ **Inc**$(n_k), \mathcal{Q}_{\text{sol}}^{k+1} \leftarrow \{\}, \mathcal{Q}_{\text{act}}^{k+1} \leftarrow$ **GetEasy**$(n_{k+1})$;
    **while** *not converge with $n_k$* **do**

        // Policy Update
        $\mathcal{M}_{\text{train}} \leftarrow$ **Sample**$(B \times z, \mathcal{Q}^k) +$ **Sample**$(B \times (1 - z), \mathcal{Q}^{k+1})$;
        Train and evaluate $\pi_\theta$ on $\mathcal{M}_{\text{train}}$ via MARL;
        // Task Expansion
        expand $\mathcal{Q}_{\text{sol}}^{\{k,k+1\}}$ and $\mathcal{Q}_{\text{act}}^{\{k,k+1\}}$ using the value estimates on $\mathcal{M}_{\text{train}}$;
        // generate novel exploration tasks
        $\mathcal{M}_{\text{seed}} \leftarrow$ active tasks from $\mathcal{M}_{\text{train}}$ that just become solved under $\pi_\theta$;
        $\mathcal{M}_{\text{exp}} \leftarrow$ generate $B_{\exp}$ exploration task samples from seeding tasks in $\mathcal{M}_{\text{seed}}$;
        Add exploration tasks $\mathcal{M}_{\text{exp}}$ to $\mathcal{Q}_{\text{act}}^{\{k,k+1\}}$;
        // Entity Progression
        Decay $z$;

    $k \leftarrow k + 1$;
**until** $n_k = N$;

---

## 3.2 Entity Progression for Massive Multi-Agent Learning

Directly applying the Stein variational update (Eq.(5)) over the relaxation $z$ of the discrete variable $n$ results in a uniform expansion over the entire $N$-dimensional space of $z$, which can be computational challenging when $N$ is particularly large. We leverage an important inductive bias in cooperative MARL that problems with more agents are generally harder than fewer agents, which implies a monotonic update scheme over the $z$ space by starting with $z_{n_0} = 1$, where $n_0$ is the smallest possible number of agents, and gradually increasing the probability mass of $z_k$ for larger $k$ values.

*Entity Progression* simplifies this procedure further by restricting $z$ to only have two non-zero entries, i.e., $z_k + z_{k'} = 1$ with $k' > k$, which suggests a smooth and progressive transition from a simpler problem with $k$ agents (i.e., $z_k = 1$) to a harder problem with $k'$ agents (i.e., $z_{k'} = 1$). Note that we can increase the agent number incrementally (i.e., $k' = k + 1$) or at an exponential rate (i.e., $k' = 2k$), the latter of which is typically more preferred in practice. We remark that entity progression can be directly applied to problems with multiple discrete parameters by viewing $k$ as a discrete vector.

## 3.3 VACL: A Hierarchical Automatic Curriculum Learning Algorithm

Since the change of discrete variable $n$ may cause a drastic distribution shift of value functions, instead of using a joint continuous representation unifying both $n$ and $\phi$, VACL combines *Entity Progression* and *Task Expansion* in a hierarchical manner: at the low level, task expansion maintains a separate curriculum $q^n(\phi)$ for each $n$ and performs particle-based curriculum update under a fixed $n$; at the high level, entity progression progressively increases $n$ to $n'$ by smoothly switching training distribution from $q^n(\phi)$ to $q^{n'}(\phi)$. The full procedure is summarized in Algo. 1.

## 3.4 Connection to Existing Works

We focus on goal-conditioned multi-agent cooperative problems and assume a paramterized task space. This is a common setting in the recent Automatic Curriculum Learning (ACL) literature [4, 25]. Some works also consider a fixed set of tasks [20], which are typically referred to as multi-task learning [33]. The core idea of ACL is to train the policy using tasks with moderate difficulty, i.e., neither too hard nor too easy, which naturally emerges as a consequence of our variational paradigm.

A popular class of ACL methods learns a generative model, such as VAE [26], GAN [9, 7] and Gaussian mixture model [24, 12, 22], over the task space with density concentrated on tasks with

desired difficulties. These methods can be viewed as a generic solution under our variational objective (Eq. 3) by representing $q(\phi)$ as a generative model and directly optimizing $\mathcal{L}_2$ whenever the policy is updated. However, it can be extremely sample-inefficient to repeatedly learn $q(\phi)$ from scratch without leveraging the fact that $q(\phi)$ is ever-expanding in cooperative MARL problems.

Another type of ACL methods is particle-based, which maintains a set of solvable task samples and gradually expands the set towards the entire space. [10, 11] assume a fixed goal and generate tasks with starting states increasingly far away from the goal; [1, 21] consider a task space over simulator configurations and train from the easiest settings to the hardest for sim-to-real adaptation. These methods are conceptually similar to the *Task Expansion* component in VACL for its particle-based nature and can be viewed as a random search procedure for optimizing $\mathcal{L}_2$ in a gradient-free manner. Our method provides a principled interpretation of curriculum updates, and achieves superior performance in practice. Besides, [30] learn an individual policy paired with each task for discovering diverse behaviors while our work learns a single goal-conditioned policy for the entire task space; [32] defines the active set as those parts of the task space in which the value function exhibits the largest gradient w.r.t. the task variable and proposes to meta-learn $f^*(\phi)$ to more effectively expand the curriculum distribution, which is complementary to our work. [14, 15] propose novel optimization methods based on the value function to minimize the KL-Divergence between $q(\phi)$ and $p(\phi)$ while our work provides a gradient-based exploration method to expand $q(\phi)$ to the entire task space.

[18, 31] propose *population curriculum*, which trains entity-invariant policies over training tasks with a growing number of agents. [8] adopt a similar curriculum learning method for Sokoban with an increasing number of boxes. These methods can be viewed as a special case of our *Entity Progression* component by applying a hard transfer from $z_k = 1$ to $z_{k'} = 1$ while our method smoothly switches between different number of entities and also yields better empirically performances.

# 4  Experiment

We consider four tasks over two environments, *Simple-Spread* and *Push-Ball* in the multi-agent particle-world environment (MPE) [19], and *Ramp-Use* and *Lock-and-Return* in the hide-and-seek environment (HnS) [2]. Every experiment is repeated over 3 seeds and performed on a desktop machine with one 64-core CPU and one 2080-Ti GPU, which is used for forward action computation and training updates. More details can be found in Appendix.

## 4.1  The Multi-Agent Particle-World Environment

The two tasks are illustrated in Fig. 3. In *Simple-Spread*, there are $n$ agents and $n$ landmarks and the agents need to occupy all the landmarks. The agents are penalized for collisions and only receive a positive reward when all the landmarks are covered. In *Push-Ball*, there are $n$ agents, $n$ balls and $n$ landmarks. The agents need to physically push the balls to get every landmark covered. A success reward is given after all landmarks are covered. We scale $n$ at an exponential rate following [18].

### 4.1.1  Main Result

We present most important results in this section and more analysis can be found in App. B. We compare *VACL* with 5 baselines, including (1) multi-agent PPO [34] with uniform task sampling (*Uniform*), (2) population curriculum only (*PC-Unif*), (3) reverse curriculum generation (*RCG*) [10], (4) automatic goal generation (*GoalGAN*) [9], which uses a GAN to generate training tasks, (5) adversarially motivated intrinsic goals (*AMIGo*) [6], which learns a teacher to generate increasingly challenging goals. We test all methods in *Simple-Spread* with $n = 8$ and *Push-Ball* with $n = 4$. For *PC-Unif* and *VACL*, we start with $n_0 = 4$ in *Simple-Spread* and $n_0 = 2$ in *Push-Ball* and then switch to the desired agent number. Evaluation is always performed on the final $N$. Results are shown in Fig. 4, where VACL outperforms all baselines with a clear margin. In *Simple-Spread*, all baselines fail to solve the task: Uniform and PC-Unif fail to gain sufficient progress due to sparse rewards; RCG quickly runs out of active tasks; GoalGAN and AMIGo takes an extremely long time to train a good neural goal-generator. In *Push-Ball*, Uniform and PC-Unif learn much slower, RCG quickly converges to a local minimum without further policy improvements, GoalGAN and AMIGo again consumes a large number of samples to make training progress.

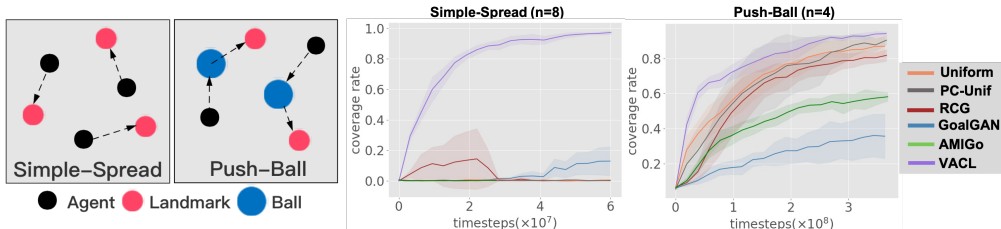

Figure 3: MPE tasks.

Figure 4: Comparison of VACL and baselines on MPE.

In addition to the baselines above, we also consider the setting of **massive agents** on *Simple-Spread* and compare our VACL method with other coverage numbers in the existing literature. We consider two existing works. The *attentional communication* (ATOC) method [13], which adopts intra-agent communication channels and trains $n = 50$ and $n = 100$ agents with *dense* rewards. ATOC is not open-sourced[4]. We re-train VACL on the same environment configuration (i.e., same agent/landmark size and room size) as ATOC for a fair comparison with the num-

Table 1: The best coverage rate ever reported on *Simple-Spread*.

| $n$ | EPC | ATOC | VACL |
|---|---|---|---|
| 24 | 56.8% | / | **97.6%** |
| 50 | / | 92% | **98.5%** |
| 100 | / | 89% | **98%** |

ber reported in [13]. The other work is evolutionary population curriculum (EPC) [18], which reports the coverage rate on the standard *Simple-Spread* task with $n = 24$ agents. EPC adopts dense rewards and separate policies for each agent while VACL trains a shared policy with *sparse 0/1 rewards*. The results are reported in Tab. 1. VACL achieves a $98\%$ coverage rate with $n = 100$ agents, which outperforms the highest number ever reported to the best of our knowledge.

#### 4.1.2 Ablation Study

**Choice of seed task:** As implied by Eq. (5) and Eq. (6), VACL explores from the boundary between $\mathcal{Q}_{\mathrm{act}}$ and $\mathcal{Q}_{\mathrm{sol}}$ by taking newly solved tasks as seeds to generate exploration tasks. A naive alternative is to simply take active tasks from $\mathcal{Q}_{\mathrm{act}}$ as seeds (*Exp.Act.*), which is similar to the exploration scheme in the RCG algorithm [10]. Note that exploration from active tasks may possibly produce unsolvable tasks, we consider an additional enhanced variant (*Exp.Act w. eval*), which take active tasks as seeds but also perform value estimates (and training) on those exploration tasks to filter out unsolvable tasks before adding them to $\mathcal{Q}_{\mathrm{act}}$. Results of these variants are shown in Fig. 5(a). Using active tasks as seeds (*Exp.Act.*) leads to a clear failure since the aggressive exploration brings too many unsolvable tasks to $\mathcal{Q}_{\mathrm{act}}$. Although additional evaluation on exploration tasks (*Exp.Act.w.eval*) significantly stabilizes training, it still yields much worse training progress than the standard VACL. This suggests that it is critical to explore from the vicinity of solved subspace.

**SVGD-Principled Update:** Our Stein variational paradigm suggests a gradient-based update rule to explore novel tasks in Sec. 3.1.3. However, most existing particle-based algorithms adopts a much simplified gradient-free alternative via uniform random noise, i.e., $\phi_{\mathrm{exp}} \leftarrow \phi_{\mathrm{seed}} + \mathrm{Unif}(-\delta, \delta)$. To illustrate the effectiveness of our SVGD-principled update rule, we consider a challenging *Hard-Spread* scenario in Fig. 5(b) with 4 agents and 4 landmarks. Obstacles are in black and landmarks can only be placed in the right side. We initialize $\mathcal{Q}_{\mathrm{act}}$ with easy tasks where each landmark has an agent nearby and evaluate the performance of the learning policy on those hardest tasks in Fig. 5(b). We remark that due to the highly constrained task space, it becomes particularly challenging for the ACL method to gradually guide the agents to learn to go through the narrow doors between obstacles. We evaluate the test performances of VACL and the simplified gradient-free version (*Unif. noise*) in Fig. 5(c). Our principled rule produces a stable learning curve with fast convergence while the gradient-free version fails to solve the task.

**Rejection Sampling:** To investigate the effectiveness of rejection-sampling-based exploration (*R.J.*), we turn off rejection-sampling (i.e., strictly following Eq ( 6)) and conduct experiments on *Hard-Spread* (i.e., highly constrained task space) and *Simple-Spread* (i.e., less constrained task space) in Tab. 2.

---

[4]We contacted the authors but did not get a response.

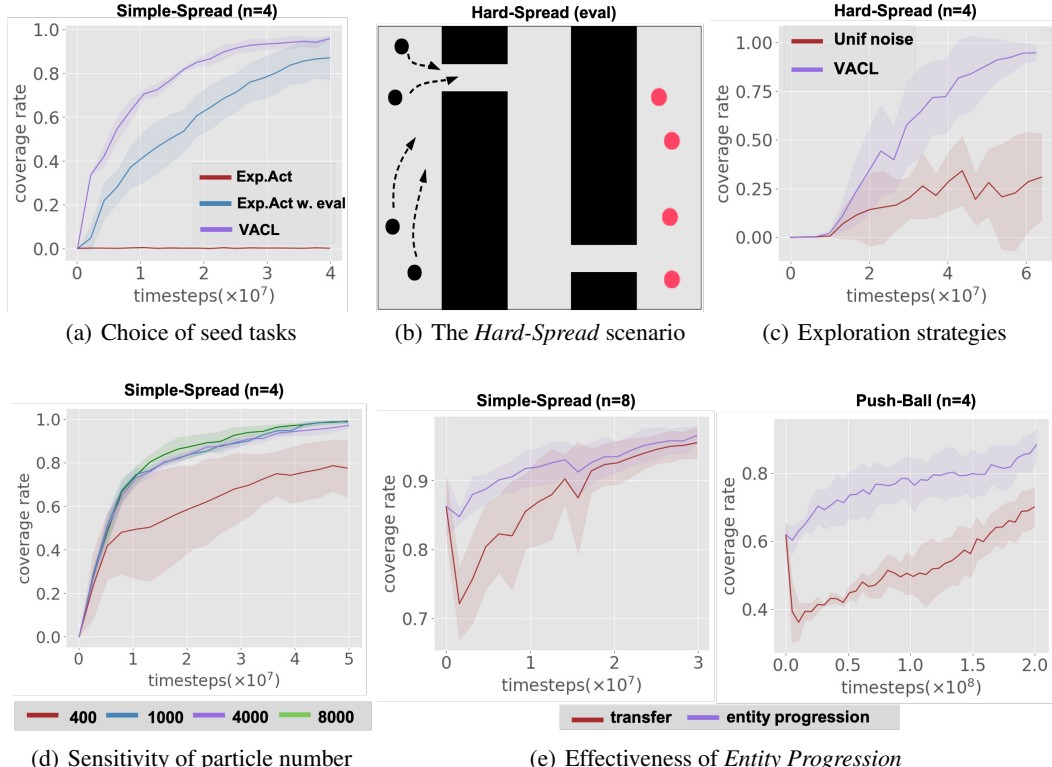

Figure 5: Ablation studies. (a) We evaluate different choices of seed tasks in *Simple-Spread*. (b) We select the hardest tasks to evaluate the performance of different exploration strategies in the *Hard-Spread* scenario, which has a highly constrained task space. (c) We compare our SVGD-principled exploration strategy with random exploration from seed tasks in *Hard-Spread*. (d) We perform experiments in *Simple-Spread* using different numbers of particles to represent $\mathcal{Q}$. (e) We compare entity progression with direct entity transfer in standard *Simple-Spread* and *Push-Ball*.

Rejection-sampling-based exploration clearly improves the final performance, particularly in *Hard-Spread*, where purely gradient-based expansion (i.e., Eq.(6)) can hardly discover feasible novel tasks. It is worth mentioning that these feasibility constraints are shared across all the experiments, so all the baselines, including RCG, GoalGAN, and AMIGo, have access to the feasibility check and are able to produce feasible training tasks.

Table 2: Ablation study on the rejection-sampling technique in Task Expansion on *Simple-Spread* and *Hard-Spread* with $n = 4$

| coverage rate % | w. R.J. | w.o. R.J. |
|---|---|---|
| *Simple-Spread* | $96.0 \pm 1.9$ | $94.1 \pm 4.9$ |
| *Hard-Spread* | $97.2 \pm 1.4$ | $0$ |

**Sensitivity of Particle Number:** We perform experiments with different numbers of particles in *Simple-Spread* with $n = 4$ and the results in Fig. 5(d) show that we do need sufficient particles to approximate $q(\phi)$. Insufficient particle size (e.g., 400) results in inaccurate estimation and poor performance. The results are robust when particle size is at least 1000. We choose 4000 particles to approximate $q(\phi)$ in all our experiments.

**Entity Progression:** We compare *entity progression* with naive population curriculum, i.e., directly transfer the trained policy on $n_{k-1}$ to the next entity size $n_k$ (*transfer*). We train with the two methods on *Simple-Spread* from $n_0 = 4$ to $N = 8$ and on *Push-Ball* from $n_0 = 2$ to $N = 4$. The results are shown in Fig. 5(e). In both two tasks, direct transfer yields a significant performance drop, which we believe due to the drastic training distribution shift. For *entity progression*, due to its smooth transition between the continuous relaxation of $z_k = 1$ and $z'_k = 1$, it produces a much stable training curves. *Entity progression* also yields the much better final performances particularly on the more challenging *Push-Ball* scenario.

## 4.2 The Hide-and-Seek Environment

We consider two tasks in the hide-and-seek environment (HnS) as shown in Fig. 6, i.e., a simplified hide-and-seek game called *Ramp-Use* and a multi-agent and sparse-reward variant of the *Lock-and-Return* task from the transfer task suite in [2]. *Ramp-Use* is a single-agent scenario for the ramp use strategy, with 1 ramps, 1 movable seeker and 1 fixed hider in the quadrant room. We need to train a seeker policy to use the ramp to get into the quadrant room for positive rewards. In *Lock-and-Return*, the discrete parameter $n$ becomes a 2-dimensional vector and each dimension represents the number of agents and boxes respective. Agents need to lock all the boxes and return to their birthplaces. Agents get a reward when *all* boxes are locked and another success reward when the task is finished, i.e., all boxes locked and all agents back to the birthplaces.

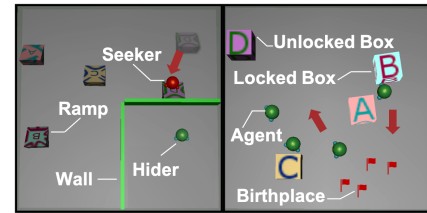

Ramp-Use    Lock-and-Return

Figure 6: Tasks in the hide-and-seek environment (HnS).

The results of different methods are shown in Tab. 3. We train each algorithm on *Ramp-Use* for 25M timesteps and on *Lock-and-Return* with $n = (2, 2)$ for 180M timesteps. For *Lock-and-Return* with $n = (4, 4)$, we train VACL for a total number of 230M timesteps. We report the average success rate over 3 seeds. VACL is the only algorithm able to solve both two tasks.

Table 3: Results of VACL and baselines in HnS tasks.

|  |  | Uniform | RCG | GoalGAN | AMIGo | VACL |
|---|---|---|---|---|---|---|
| Ramp-Use | $n = 1$ | $15.3\% \pm 16.1\%$ | $76.7\% \pm 2.1\%$ | $6.4\% \pm 7.5\%$ | $40.8\% \pm 29.2\%$ | $99.7\% \pm 0.5\%$ |
| Lock-and-Return | $n = (2, 2)$ | $<1\%$ | $5.0\% \pm 5.1\%$ | $<1\%$ | $< 2\%$ | $97.3\% \pm 0.1\%$ |
|  | $n = (4, 4)$ | / | / | / | / | $97.0\% \pm 1.6\%$ |

In *Lock-and-Return*, we start with the easy setting of $n_0 = (2, 2)$ and proceed towards $N = (4, 4)$. Note that $n$ is a 2-dimensional discrete vector in this task, there are multiple paths to transit from the continuous relaxation $z_{(2,2)} = 1$ to $z_{(4,4)} = 1$. One is to double the number of agents first and keep the box number unchanged, and then double the number of boxes, i.e., $z_{n_0} = 1 \rightarrow z_{(4,2)} = 1 \rightarrow z_N = 1$ (*agent-first*). The other is to first double the number of boxes and then increase the agent number, i.e., $z_{n_0} = 1 \rightarrow z_{(2,4)} = 1 \rightarrow z_N = 1$ (*box-first*). Although both two paths are feasible transitions, they lead to different practical performances. The results are shown in Fig. 7, where we clearly observe that the strategy of increasing agent number first substantially outperforms the box-first strategy. Therefore, we generally suggest to increase the number of agents before objects in entity progression. We remark that since no baseline achieves a success rate above 15% even in the easy case of $n = (2, 2)$, we do not evaluate them in the hard setting.

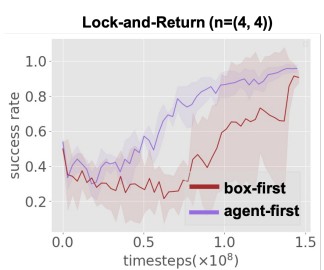

Figure 7: Results of different entity switching strategies.

## 5 Conclusion

In this paper, we propose a novel curriculum learning paradigm following the principle of Stein variational inference and develop an effective algorithm, Varitional Automatic Curriculum Learning (VACL), which solves a collection of sparse-reward multi-agent cooperative problems. In this work, we consider cooperative tasks for the clear measurement of training progress. We also remark that VACL assumes a monotonic expansion of $q(\phi)$, which only holds in the fully cooperative setting. There are also existing works that study ACL for competitive games [28, 16, 2], which we leave as future work. Our framework tackles an important research problem in the literature while experiments

are conducted on open-source environments with student licences, so we believe our work does not produce any negative impact to the society.

## Acknowledgements

This work is supported by National Science Foundation of China (No. U20A20334, U19B2019, 61832007), National Key R&D Program of China (No.2019YFF0301505), Tsinghua EE Xilinx AI Research Fund and 2030 Innovation Megaprojects of China (Programme on New Generation Artificial Intelligence) Grant No. 2021AAA0150000. We appreciate Jiantao Qiu and Xuefei Ning for their reviews of an early draft. We would like to thank Chao Yu for her support and input during this project. Yi Wu would also thank Bowen Baker and Ingmar Kanitscheider since some of the very initial ideas are inspired by their early discussion at OpenAI. Finally, we thank all anonymous reviewers for their constructive comments.

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
