# Supplementary material for Variational Automatic Curriculum Learning for Sparse-Reward Cooperative Multi-Agent Problems

**Jiayu Chen** [1♯], **Yuanxin Zhang**[1], **Yuanfan Xu**[1], **Huimin Ma**[3],
**Huazhong Yang**[1], **Jiaming Song**[4], **Yu Wang**[1], **Yi Wu**[12♮]
[1] Tsinghua University, [2] Shanghai Qi Zhi Institute,
[3] University of Science and Technology Beijing, [4] Stanford University,
♯jia768167535@gmail.com, ♮jxwuyi@gmail.com

All the source code can be found at our project website `https://sites.google.com/view/vacl-neurips-2021`.

## A  Proofs

In order to prove Theorem 1, we introduce the following lemma, which uses Assumption 1.

**Lemma 1.** *Let $T(\phi) = \phi + \epsilon f(\phi)$ and $q_{[T]}(\psi)$ the density of $\psi = T(\phi)$. Then*

$$\nabla_\epsilon \mathcal{L}_2(\pi, q_{[T]})|_{\epsilon=0} = -\mathbb{E}_{\phi \sim q(\phi)}[V(\psi, \pi) \cdot \mathrm{trace}(\nabla_\phi \log p(\phi) f(\phi)^\top + \nabla_\phi f(\phi))] \quad (7)$$

*with $\mathcal{L}_2(\phi, \pi)$ as defined in Eq.(3).*

*Proof.* The proof is largely based on [2]. Denote by $p_{T^{-1}}(\phi)$ the density of $\phi = T^{-1}(\psi)$ when $\psi \sim p(\psi)$, then:

$$\mathcal{L}_2(\pi, q_{[T]}) = \mathbb{E}_{\psi \sim q_{[T]}(\psi)}\left[V(\psi, \pi) \log \frac{p(\psi)}{q_{[T]}(\psi)}\right] = \mathbb{E}_{\phi \sim q(\phi)}\left[V(T(\phi), \pi) \log \frac{p_{[T^{-1}]}(\phi)}{q(\phi)}\right] \quad (8)$$

and since $\epsilon$ only depends on $T$, we have:

$$\nabla_\epsilon \mathcal{L}_2(\pi, q_{[T]}) = \mathbb{E}_{\phi \sim q(\phi)}\left[V(T(\phi), \pi)\nabla_\epsilon \log p_{[T^{-1}]}(\phi) + \log \frac{p_{[T^{-1}]}(\phi)}{q(\phi)}\nabla_\epsilon V(T(\phi), \pi)\right] \quad (9)$$

From Assumption 1, $V(\phi', \pi)$ is constant within a small vicinity of $\phi$; thus $\nabla_\epsilon V(T(\phi), \pi)|_{\epsilon=0} = 0$; hence

$$\nabla_\epsilon \mathcal{L}_2(\pi, q_{[T]})|_{\epsilon=0} = \mathbb{E}_{\phi \sim q(\phi)}\left[V(T(\phi), \pi) \cdot \nabla_\epsilon \log p_{[T^{-1}]}(\phi)|_{\epsilon=0}\right] \quad (10)$$

Define $s_p(\phi) = \nabla_\phi \log p(\phi)$; we have

$$\nabla_\epsilon \log p_{[T^{-1}]}(\phi) = s_p(T(\phi))^\top \nabla_\epsilon T(\phi) + \mathrm{trace}((\nabla_\phi T(\phi))^{-1} \cdot \nabla_\epsilon \nabla_\phi T(\phi)). \quad (11)$$

When $T(\phi) = \phi + \epsilon f(\phi)$ and $\epsilon = 0$, we have

$$T(\phi) = \phi, \quad \nabla_\epsilon T(\phi) = f(\phi), \quad \nabla_\phi T(\phi) = I, \quad \nabla_\epsilon \nabla_\phi T(\phi) = \nabla_\phi f(\phi), \quad (12)$$

where $I$ is the identity matrix. Therefore:

$$\nabla_\epsilon \mathcal{L}_2(\pi, q_{[T]})|_{\epsilon=0} = \mathbb{E}_{\phi \sim q(\phi)}\left[V(T(\phi), \pi)(\nabla_\phi \log p(\phi)^\top f(\phi) + \mathrm{trace}(\nabla_\phi f(\phi)))\right] \quad (13)$$

$$= \mathbb{E}_{\phi \sim q(\phi)}\left[V(T(\phi), \pi) \cdot \mathrm{trace}(\nabla_\phi \log p(\phi) f(\phi)^\top + \nabla_\phi f(\phi))\right] \quad (14)$$

where the final equality represents an inner product with a trace. □

35th Conference on Neural Information Processing Systems (NeurIPS 2021).

**Theorem 1.** *Let $T(\phi) = \phi + \epsilon f(\phi)$ where $f$ is an element of some vector-valued RKHS of a positive definite kernel $k(\phi, \phi') : \Phi \times \Phi \to \mathbb{R}$, and $q_{[T]}$ the density of $\psi = T(\phi)$ when $\phi \sim q$, then*

$$\nabla_f \mathcal{L}_2(q_{[T]}, \pi)|_{f=0} = f^*(\phi), \tag{15}$$

*where $f^*(\cdot) = \mathbb{E}_{\phi' \in \mathcal{Q}}[V(\phi', \pi)(k(\phi', \cdot)\nabla_{\phi'} \log p(\phi') + \nabla_{\phi'} k(\phi', \cdot))]$.*

*Proof.* Let $\mathcal{H}^d = \mathcal{H} \times \cdots \times \mathcal{H}$ be a vector-valued RKHS, and $F[f]$ be a functional of $f$. The gradient $\nabla_f F[f]$ of $F[\cdot]$ is a function in $\mathcal{H}^d$ that satisfies

$$F[f + \epsilon g] = F[f] + \epsilon \langle \nabla_f F[f], g \rangle_{\mathcal{H}^d} + O(\epsilon^2). \tag{16}$$

Define $F[f] = \mathbb{E}_{\phi \sim q(\phi)} \left[ V(\phi + f(\phi), \pi) \left( \log p_{[(\phi + f(\phi))^{-1}]}(\phi) - \log q(\phi) \right) \right]$, we have

$$F[f + \epsilon g] = \mathbb{E}_{\phi \sim q(\phi)} \left[ V(\phi + f(\phi) + \epsilon g(\phi), \pi) \cdot \left( \log p_{[(\phi + f(\phi) + \epsilon g(\phi))^{-1}]}(\phi) - \log q(\phi) \right) \right] \tag{17}$$

$$= \mathbb{E}_{\phi \sim q(\phi)} \left[ V(\phi + f(\phi), \pi) \cdot \left( \log p(\phi + f(\phi) + \epsilon g(\phi)) - \log q(\phi) \right. \right. \tag{18}$$
$$\left. \left. + \log \det(I + \nabla_\phi f(\phi) + \epsilon \nabla_\phi g(\phi)) \right) \right]$$

where $V(\phi + f(\phi), \pi) = V(\phi + f(\phi) + \epsilon g(\phi), \pi)$ for $f \approx 0, \epsilon \approx 0$ from Assumption 1.

Then, we have that:

$$\mathbb{E}_q[V(\phi + f(\phi), \pi) \cdot \left( \log p(\phi + f(\phi) + \epsilon g(\phi)) - \log p(\phi + f(\phi)) \right)] \tag{19}$$

$$= \epsilon \cdot \mathbb{E}_q[V(\phi + f(\phi), \pi) \cdot \nabla_\phi \log p(\phi + f(\phi)) \cdot g(\phi)] + O(\epsilon^2) \tag{20}$$

$$= \epsilon \cdot \mathbb{E}_q[V(\phi + f(\phi), \pi) \cdot \nabla_\phi \log p(\phi + f(\phi)) \cdot \langle k(\phi, \cdot), g \rangle_{\mathcal{H}^d}] + O(\epsilon^2) \tag{21}$$

$$= \epsilon \cdot \langle \mathbb{E}_q[V(\phi + f(\phi), \pi) \cdot \nabla_\phi \log p(\phi + f(\phi)) \cdot k(\phi, \cdot), g \rangle_{\mathcal{H}^d}] + O(\epsilon^2), \tag{22}$$

where the first equality uses the definition of functional gradient and second equality uses the representation theorem for RKHS. Similarly, we also have that:

$$\mathbb{E}_q[V(\phi + f(\phi), \pi) \cdot \left( \log \det(I + \nabla_\phi f(\phi) + \epsilon \nabla_\phi g(\phi)) - \log \det(I + \nabla_\phi f(\phi)) \right)] \tag{23}$$

$$= \epsilon \langle \mathbb{E}_q[V(\phi + f(\phi), \pi) \cdot \text{trace}((I + \nabla_\phi f(\phi))^{-1} \cdot \nabla_\phi k(\phi, \cdot))], g \rangle + O(\epsilon^2). \tag{24}$$

Therefore, by combining Equation (22) and Equation (24), we have that

$$F[f + \epsilon g] = F[f] + \epsilon \langle \nabla_f F[f], g \rangle_{\mathcal{H}^d} + O(\epsilon^2), \tag{25}$$

where

$$\nabla_f F[f] = \mathbb{E}_{\phi \in \mathcal{Q}}[V(\phi, \pi) \cdot (k(\phi, \cdot)\nabla_\phi \log p(\phi + f(\phi)) + \text{trace}((I + \nabla_\phi f(\phi))^{-1} \cdot \nabla_\phi k(\phi, \cdot))],$$

and taking $f = 0$ completes the proof. $\qquad\square$

## B    Additional Results

**Pure Task Expansion Results on MPE:** VACL contains entity progression in the result of Fig. 4. To specifically study the performance of task expansion, we exclude entity progression module from VACL and compare with baselines in *Simple-Spread* with $n = 4$ and *Push-Ball* with $n = 2$. For a fair comparison, we also provide additional experiments to combine GoalGAN and AMIGo with the initial knowledge of easy tasks. As show in Fig. 8, VACL without entity progression also outperforms all the baselines in the two environments.

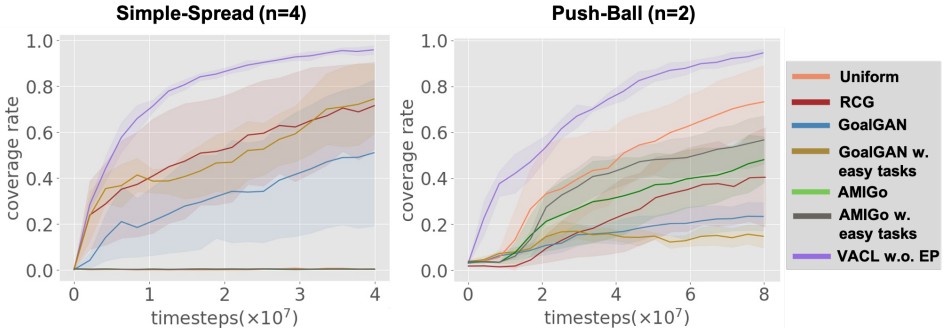

Figure 8: Comparison of baselines and VACL without entity progression on MPE (i.e., task expansion only).

**Additional Results on SVGD-Principled Update:** We additionally conduct experiments to compare VACL with the gradient-free version (*Unif. noise*) in the original *Simple-Spread* with $n = 4$ and *Push-Ball* with $n = 2$. As shown in Fig. 9, VACL is comparable with the variant using gradient-free exploration in *Simple-Spread* and the gap becomes significant in *Push-Ball*, which implies a gradient-based update rule can explore more novel tasks than the simplified gradient-free method in harder scenarios, which is consistent with what we find in the main paper.

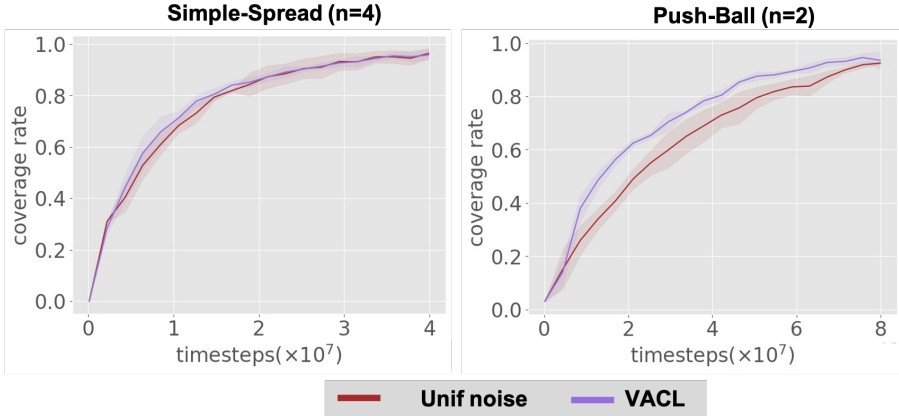

Figure 9: Comparison of VACL and the gradient-free exploration method in *Simple-Spread* and *Push-Ball*

**VACL for Heterogeneous Agents:** We add another experiment domain with heterogeneous agents in *Speaker-Listener* (Fig. 10(a)), which is one of the basic tasks in the MADDPG [4] paper. This task consists of two cooperative agents, a speaker and a listener, and three landmarks with different colors. The speaker and listener obtain +1 reward when the listener covers the correct landmark. However, while the listener can observe the relative position and color of the landmarks, it does not know which landmark it should navigate to. Conversely, the speaker's observation consists of the correct landmark color, producing a communication output that the listener can observe. Thus, the speaker must learn to output the landmark color based on the motions of the listener. The results in Fig. 10(b) show that our algorithm can also work on the problems with heterogeneous agents.

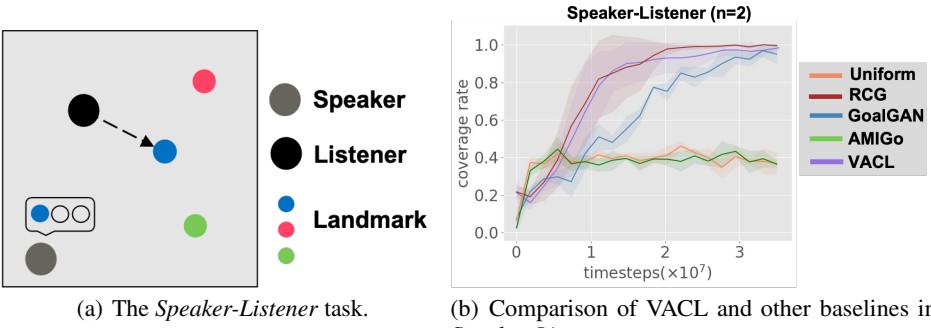

(a) The *Speaker-Listener* task.     (b) Comparison of VACL and other baselines in *Speaker-Listener*.

Figure 10: We compare VACL with other baselines in *Speaker-Listener*. The results show that our algorithm can also handle the problems with heterogeneous agents

## C Environment Details

### C.1 Environment Configurations

In *Simple-Spread*, agents get +4 reward when all the landmarks are occupied and get a reward of -1 when any agents collide with each other. The agents are only penalized by collision once at every timestep. In addition, agents and landmarks are randomly generated in a square area with a side length of 6 in the evaluation process. This is the same task as the Cooperative Navigation game in the MADDPG paper [4]. We generate entities in a larger map to increase the difficulty. A larger scene means it is more difficult for agents to get positive reward signals in the sparse-rewards setting. We construct the *Hard-Spread* scenario by adding walls to separate the room into three parts. Agents can observe the position of landmarks, but the walls are invisible. The *Hard-Spread* mentioned in Fig. 5(b) is a $10 \times 2$ rectangle.

In *Push-Ball*, there are $n$ agents, $n$ balls and, $n$ landmarks. Agents will get a shared reward $2/n$ per timestep when any ball occupies one landmark. If all of the landmarks are occupied, agents will get an extra +1 reward. The collision penalty is the same as *Simple-Spread*. Entities are randomly generated in a $4 \times 4$ square area in the evaluation process. Push-Ball is our first proposed sparse-rewards task based on the physical kernel of the particle world in the MADDPG [4] paper.

For the tasks in the particle-world environment, we evaluate the performances of our algorithm and baselines with the average coverage of landmarks in the last five evaluation steps within every episode.

In *Lock-and-Return*, if all the boxes are locked, agents get +0.2 reward. Moreover, agents can get a success bonus of +1 if they return to their birthplaces after all the boxes are locked. We test our algorithm and baselines on a floor size of 12, with sides twice as the standard hide-and-seek environment. The environment is fully-observable in our setting for simplicity. In this task, we adopt the mean return rate of agents for comparison in the last five evaluation steps.

In *Ramp-Use*, we hope the seeker to learn how to use ramps to enter the enclosed room and catch the hider (Fig. 6). When the hider is spotted, the seeker gets a reward of +1. Otherwise, he gets -1. The environment is fully-observable, and the hider is fixed in the room. We evaluate the performance with the success rate of finding the hider in the last five steps.

### C.2 Definition of $\phi$ and GetEasy(n)

$\phi$ is a vector that contains positions of agents and landmarks in *Simple-Spread*, and positions of agents, balls, and landmarks in *Push-Ball*. In *Lock-and-Return*, it contains positions of agents and boxes (without birthplaces). In *Ramp-Use*, we concatenate positions of the hider, boxes, and the ramp to get $\phi$. As for easy cases that generated by **GetEasy**(n), we consider those cases where agents are near landmarks in *Simple-Spread*, and agents, balls, and landmarks are close to each other in *Push-Ball* as easy tasks. In *Lock-and-Return*, easy cases have agents near birthplaces, and boxes randomly placed near them. In *Ramp-Use*, easy cases have the ramp right next to the wall, and agents located next to the ramp.

## D Training Details

### D.1 The Multi-Agent Particle-World Environment

In the two particle-world environments, *Simple-Spread* and *Push-Ball*, we use the same network architecture with a self-attention mechanism (Fig. 11) as EPC [3]. The value network is divided into two parts, which accepts observations of all the agents as input and output the V-value *V*. The first part (right in Fig. 11), named *observation encoder*, is used to encode the observation of agent *j* into $f_i(o_j)$. For each agent *j*, *observation encoder* takes observation $o_j$ as input and then applies an entity encoder for each entity type to obtain embedding vectors of all the entities within this type. We attend the entity embedding of agent *j* with all the entities of this type to obtain an attended embedding vector. Then we concatenate all the entity embedding vectors. The observation of agent *j*, $o_{j,j}$, is directly forwarded to a fully connected layer. Finally, both the concatenated vectors and the embedded $o_{j,j}$ are forwarded to a fully connected layer to generate the output, $f_i(o_j)$. The output size of entity encoder(green and purple in Fig. 11) and attention layer (orange in Fig. 11) is 64. The

second part (left in Fig. 11) accepts the encoding vector $f_i$ of all the agents. We attend all the agent embedding $f_i(o_j)$ to the global attention embedding denoted as $g_i$ (the orange box in the left part in Fig. 11). The ith agent's own observation is forwarded to a 1-layer fully connected network to get $m_i$. The final layer is a 2-layer fully connected network that takes the concatenation of $m_i$ and the global attention embedding $g_i$ and outputs the final V value. The policy network has a similar structure as the *observation encoder* $f_i(o_i)$, which uses an attention module over the entities of each type in the observation $o_i$. We do not share parameters between the policy and value network.

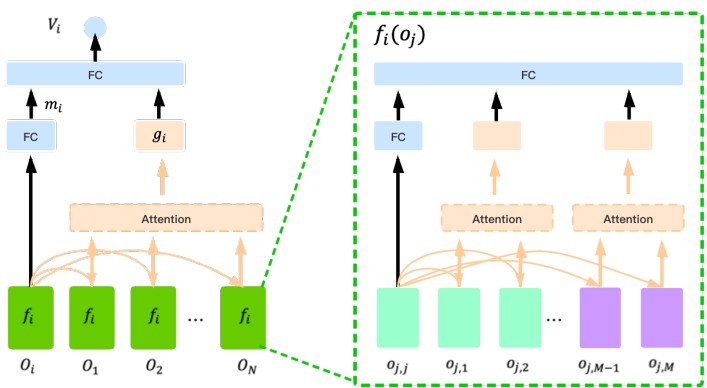

Figure 11: Our population-invariant architecture for the value function.

Agents in *Simple-Spread* and *Push-Ball* are trained by PPO. It is worth mentioning that mini-batch size should be changed when the entity size switches from $n_{k-1}$ to $n_k$ due to memory requirements. In *Simple-Spread*, we divide each batch into 2 mini-batch for $n = 4$ and 16 for $n = 8$. In *Push-Ball*, the number is 2 for $n = 2$ and 8 for $n = 4$. In addition, easy tasks are different for each training phase. We use the side length $s$ to represent the tasks generated by **GetEasy**$(n_k)$. Entities in the initial tasks are randomly generated in a $s \times s$ square region. We choose $s = 0.6$ for $n = 4$ and $s = 2.0$ for $n = 8$ in *Simple-Spread* and *Hard-Spread*. And the side length in *Push-Ball* is $s = 0.8$ for $n = 2$ and $s = 1.6$ for $n = 4$. Moreover, we define a threshold of convergence. When the evaluation result of the current policy reaches 0.9, we start entity progression. More hyper-parameter settings are shown in Tab. 4.

Table 4: VACL hyper-parameters used in the particle-world environment.

| Hyper-parameters | Value |
|---|---|
| Learning rate | 5e-4 |
| Discount rate ($\gamma$) | 0.99 |
| GAE parameter ($\lambda$) | 0.95 |
| Adam stepsize | 1e-5 |
| Value loss coefficient | 1 |
| Entropy coefficient | 0.01 |
| PPO clipping | 0.2 |
| Parallel threads | 500 |
| PPO epochs | 15 |
| Reward scale parameter | 0.1 |
| Horizon | 70 (*Simple-Spread*), 120 (*Push-Ball*) |
| Gradient step ($\epsilon$) | 0.6(*Simple-Spread*), 0.4 (*Push-Ball*) |
| Uniform noise ($\delta$) | 0.6(*Simple-Spread*), 0.4 (*Push-Ball*) |
| RBF kernel ($h$) | 1 |
| $B_{\text{exp}}$ | 150 |

## D.2   The Hide-and-Seek Environment

In the hide-and-seek environment, we use the same attention mechanism as the architecture in [5] for the policy and critic network. In addition, we train a recurrent policy and split data into chunks in HnS

which is similar to Bowen Baker et al. [1]. The size of hidden states is 64. In *Lock-and-Return*, the initialization tasks in the active set are defined in the quadrant room where agents are near the boxes. In *Ramp-Use*, the ramp is placed against the wall, and the seeker is near the ramp for initialization. It is worth mentioning that we divide the floor into grids and define gradient step $\epsilon$ and uniform noise $\delta$ with the number of grids in Hns. The grids are $60 \times 60$ in *Lock-and-Return* and $30 \times 30$ in *Ramp-Use*. We set $\epsilon$ and $\delta$ to 1 in the two environments. More hyper-parameter settings are shown in Tab. 5.

Table 5: VACL hyper-parameters used in the hide-and-seek environment.

| Hyper-parameters | Value |
|---|---|
| Learning rate | 5e-4 |
| Discount rate ($\gamma$) | 0.99 |
| GAE parameter ($\lambda$) | 0.95 |
| Adam stepsize ($\epsilon$) | 1e-5 |
| Value loss coefficient | 1 |
| Entropy coefficient | 0.01 |
| PPO clipping | 0.2 |
| Parallel threads | 300 |
| PPO epochs | 15 |
| Chunk length | 40 (*Lock-and-Return*),10 (*Ramp-Use*) |
| Mini-batch size | 1(*Lock-and-Return*),2 (*Ramp-Use*) |
| Horizon | 60 |
| RBF kernel ($h$) | 1 |
| $B_{\text{exp}}$ | 200 |

### D.3 Additional Implementation Details

**Diversified Queue:** We use two fixed-size queues to maintain $\mathcal{Q}_{\text{act}}$ and $\mathcal{Q}_{\text{sol}}$ for computation and memory efficiency. Note that SVGD naturally suggests a diversified data queue, which is our implementation by default. Concretely, We define a distance measure for a task $\phi$ and a task set $\mathcal{Q}$ by the average top-$k$ minimum distance between $\phi$ and each element in $\mathcal{Q}$, i.e.,

$$D(\phi, \mathcal{Q}) = \text{mean} \left( \min \left( k; \{\|\phi - \phi'\| : \phi' \in \mathcal{Q}\} \right) \right),$$

where $k = 5$ in this paper. Then when the size of $\mathcal{Q}$ exceeds the bound $L$, we remove samples w.r.t. their intra-set distance measures:

$$\mathcal{Q} \leftarrow \mathcal{Q} \setminus \min \left( |\mathcal{Q}| - L; \{D(\phi_j, \mathcal{Q}) : \phi_j \in \mathcal{Q}\} \right),$$

We set the queue capacity $L$ to 2000 for both $\mathcal{Q}_{\text{act}}$ and $\mathcal{Q}_{\text{sol}}$. Note that this greedy strategy only works when the overflow size $|\mathcal{Q}| - L$ is small, e.g., in our setting. When a large number of samples overflows, the distances need to be synced during the deletion process. Finally, we also use standard FIFO queue and don't notice any significant performance drop even using a FIFO queue, which can potentially be another practical alternative for even better computational efficiency.

**The Ratio of Active and Solved Tasks:** We follow the convention of curriculum learning to sample more training tasks from $\mathcal{Q}_{\text{act}}$ for effective training. The ratio of active and solved tasks that we use in the training batch is 95% to 5%. We also compare VACL with a baseline version with uniform sampling from $\mathcal{Q}_{\text{act}}$ and $\mathcal{Q}_{\text{sol}}$ (Uniform sampling) in *Simple-Spread* with $n = 4$ and *Push-Ball* with $n = 2$. The results are shown in Fig. 12, where VACL performs slight better which is also consistent with the principle of curriculum learning.

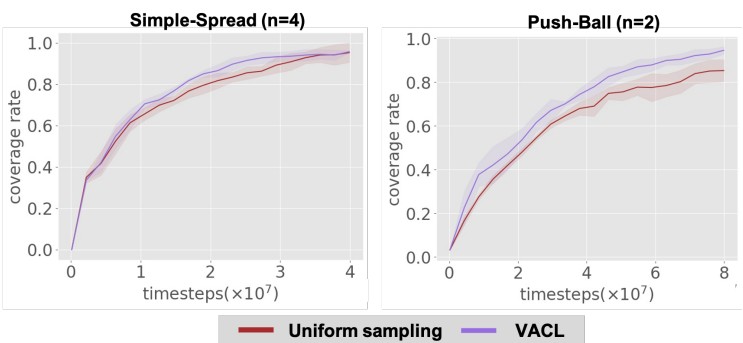

Figure 12: Comparison of VACL and the uniform sampling method in *Simple-Spread* and *Push-Ball*