# OpenReview forum: "Variational Automatic Curriculum Learning for Sparse-Reward Cooperative Multi-Agent Problems"
_NeurIPS.cc/2021/Conference — NeurIPS 2021 Poster_

### Official Review · Reviewer_spUJ · 2021-07-07

**Rating:** 6
**Confidence:** 4

**Summary:**

The paper proposes a curriculum learning method for cooperative multi-agent reinforcement learning in sparse reward tasks. The authors derive a curriculum learning approach by forming a lower bound on the expected reward under a target distribution p(\phi). This lower bound is maximized by alternatingly optimizing the policy under a variational distribution q(\phi) and then adjusting the variational distribution based on the current performance of the agent. The authors investigate the theoretical behavior showing that the optimization of q(\phi) minimizes a form of "weighted" KL-Divergence to p(\phi). Further, they derive a gradient-based approach to optimize q(\phi) and propose an approximate implementation of their approach that performs empirically well in different multi-agent RL benchmarks.

**Limitations And Societal Impact:**

The authors discuss the limitations and societal impact of their work in the conclusion.

**Main Review:**

Strengths: \
The paper presents novel and insightful content. I especially like the transition from the derived gradient in Eq (4) to the approximate implementation in Section 3.1. The proposed quantization nicely connects concepts such as success buffers to the lower bound on the agent performance. As such concepts are rarely connected to any theory, this is an exciting variation.

The experiments demonstrate good performance. In combination with the convincing theoretical motivation, this seems to be a promising approach for curriculum learning in (multi-agent) RL. Further, the ablations in the experimental section give a good impression of the inner workings of the method.

Weaknesses: \
The theoretical motivation for the curriculum does not seem to carry over to the discrete variables describing the number of agents. The introduced entity progression scheme seems detached from the theory. While I understand that the gradient-based operator may not be computationally advisable for large N, why did the authors not subsume the number of agents into the parameter \phi. While this may not be in line with the continuity assumption about \phi, it should be fairly straightforward to implement. And with a properly adjusted noise process for exploration, it could yield an implementation more similar to the theoretical motivation. Nonetheless, the overall method seems to work well.

I see another weakness in the performance comparison in the experimental section. Apart from issues with clarity (see next section), the comparison methods did not use a curriculum over the number of agents. While I understand that there may be no related method ideally suited (because of the lack of a curriculum over both goal-state and agent number), I think one ablation should be added to the experiments: VACL without the curriculum over the number of agents. This would give an additional insight into the importance of the curriculum over agents and also yield a fairer comparison to the other curriculum learning methods like GoalGAN, RCG and AMIGo.

In the "SVGD-Principled Update" experiment, the (principled) gradient-based version from Eq (6) was not compared to the rejection sampling method used in Algorithm 1. Instead, only a version without the rejection step has been used. Why is there no such comparison in the hard-spread scenario? I think it would give a good intuition of whether the rejection sampling used in VACL performs similarly to the gradient-based method.

Clarity: \
The paper is well written, and the concepts were understandable. However, some parts of the paper could be more concise, especially w.r.t. experimental details:
 - The rejection sampling makes use of an "oracle" that can identify infeasible regions. However, the infeasible regions are not specified in the experimental section except for hard-spread, where (as already mentioned) rejection sampling is not evaluated. The authors should clearly state whether additional knowledge was gained through the aforementionend oracle in the other environments. This would allow readers to judge whether VACL had an advantage over other methods like GoalGAN due to the access to the oracle.
 - Was GoalGAN also given access to the initial samples obtained by GetEasy(n)? I think it should be because it explicitly supports such seeding by initial tasks. I am curious because, in line 270, the authors mention that "GoalGAN and AMIGo takes an extremely long time to train a good neural goal-generator."
 - In Algorithm 1: A reference to the implementation of the "Sample" function in the appendix would be good to guide the reader.
 - How were the continuous goals \phi defined in the individual experiments? Unfortunately, I could not find any details on this. Especially in the hide-and-seek environment, this would be interesting. Further, what seed tasks are generated by the GetEasy function in the environments?
 - The discussion of the "Choice of seed task" in Section 4.1.2 seems to have an error. The text states that "Exp.Act.w.eval" works better than "Exp.Act." while the plots indicate the opposite.

I think the aforementioned points are another weakness of the paper, as some of those make the obtained results a bit less convincing.

Related Work: \
I have two suggestions for improvement for the related work:
 - The reference to the work by Wöhlke et al. (citation [30] in the main paper) seems to be presented a bit misleading in the related work section. Is it maybe because of a mix-up in the citations? While the authors say that "[30] propose to meta-learn f^*(\phi) to more effectively expand the curriculum distribution", it instead seems that [30] use a different characterization of the "active set" as the authors proposed here. In [30], the active set is defined as those parts of the task space in which the value function exhibits the largest gradient w.r.t. the task variable.
 - There exists work [1,2] that proposes a similar curriculum scheme and links it to a probabilistic inference perspective. It also minimizes the KL-Divergence between q(\phi) and p(\phi). While I don't think that this existing work hinders the novelty and originality of the proposed approach due to the different methodology, I think it should be included in the related work section. In combination with the corrected description of reference [30], this would yield a better overview of methods based on value functions and/or probabilistic inference.

[1] Klink, Pascal, et al. "Self-Paced Deep Reinforcement Learning." Neural Information Processing Systems, 2020.

[2] Klink, Pascal, et al. "Self-paced Contextual Reinforcement Learning." Conference on Robot Learning, 2019.


UPDATE:

After reading the authors responses, I increased my score from 5 to 6.

**Time Spent Reviewing:**

4

---

> ### Author Response · Authors · 2021-08-09
> **We will revise our paper accordingly.**
>
> Thanks for the valuable comments. We promise to add all the clarifications to our final version.
>
> 1. “theoretical motivation does not carry over to the discrete variables”
>
> This is a great question that we should have explained in the paper. It is really tricky to combine discrete variable $n$ and continuous parameter $\phi$ together since the dimension of $\phi$ is dependent on $n$: $\phi$ contains the initial configurations of all the agents (entities) in the environment and different $n$ leads to $\phi$ with different dimensions. It is definitely possible to augment the dimensions of $\phi$ with dummy dimensions to support the maximum possible number of agents in the environment, which, however, can be particularly computationally inefficient when $n$ is large (e.g., we run experiments with $n=100$ agents). Section 2.3 also explains a continuous relaxation to make discrete variables nicely fit into the continuous task parameterization assumption. However, we suffer from the same task-space-dimensionality issue. Therefore, we alternatively present a hierarchical framework with discrete variables in the outer loop so that the inner loop will have a fixed task-space dimension. Finally, we remark that, if we can have an entity-size-invariant task space representation, we can naturally combine section 2.3 and section 2.2 into a unified framework, which we will leave as a future direction.
>
> 2. “the comparison methods did not use a curriculum over the number of agents”
>
> We have already conducted this ablation study in Appendix B. To specifically study the performance of task expansion, we exclude the entity progression module from VACL and compare it with baselines in Fig 10 in the appendix. VACL without entity progression still outperforms all the baselines in Simple-Spread and Push-Ball. In addition, in Table 2 in the main paper, VACL does not involve entity progression at all for the Ramp-Use task (there is only a single agent) and Lock-and-Return with $n=(2,2)$ ($n_0$ is set to be $(2,2)$) and still significantly outperforms all the baselines. Moreover, we also compare VACL with EPC in Table 1, a population curriculum method for scaling MARL, which indeed performs a curriculum over the number of agents. Our method still performs much better than the numbers of EPC cited from its original paper.
>
> 3. “Eq (6) was not compared to the rejection sampling method”
>
> We want to clarify that the rejection sampling technique is the _default_ _choice_ in VACL and we use it for _all_ the experiments (please refer to our clarification in the next paragraph). Rejection sampling is not an alternative to Eq (6), instead, it is a fix to Eq (6) in the case of constrained task space. It is very common that the task space is constrained, e.g., the initial position of agents cannot be outside the environment boundary or collide with obstacles. Eq (6), which follows SVGD, may easily produce infeasible task configurations and cannot be directly applied to most MARL scenarios. In order to effectively apply Eq (6) to MARL settings, we introduce the rejection sampling technique as an efficient approximation to the projected gradient method, which works well in all our experiments.
> To better show the importance of rejection sampling exploration, we turn off rejection-sampling-based exploration (i.e., strictly following Eq (6)), and conduct ablation studies on Hard-Spread (i.e., highly constrained space) and Simple-Spread (i.e., less constrained task space) in the following table.
>
> | | | |
> |:-|:-:|:-:|
> |coverage rate|w. rejection-sampling-exp.|w.o. rejection-sampling-exp.|
> |Simple-Spread ($n=4$) %| $96.0\pm1.9$|$94.1\pm4.9$|
> |Hard-Spread ($n=4$) %|$97.2\pm1.4$ | $0$ |
> | | | |
>
> Rejection-sampling-based exploration clearly improves the final performance, particularly in Hard-Spread, where pure gradient expansion (Eq.(6)) can hardly discover feasible novel tasks.
>
> 4. Clarity. "rejection sampling makes use of an "oracle""
>
> We apologize for the confusion. In all the experiments, agents, objects, and obstacles (e.g., walls) cannot overlap nor be placed outside the game area. For example, in Simple-Spread, agents and landmarks are randomly generated in a predefined square region. So every environment has a constrained task space. We will clarify this in the final paper.
> Such an environmental constraint is very common in popular MARL environments and the simulator will directly determine whether the specified configuration is feasible or not. This “feasibility oracle” is shared across all the experiments, so all the baselines, including RCG, GoalGAN, and AMIGo, have access to the same oracle to produce feasible training tasks.
>
> 5. Clarity: “Was GoalGAN given access to GetEasy(n)?”
>
> GoalGAN does not require access to easy tasks. So it is theoretically more general but requires substantially more computation. Our method requires the knowledge of GetEasy(n) but is much more efficient. In the following table, we provide additional experiments and find that our algorithm still outperforms other methods with a clear margin even when the easy tasks are provided for training the task generator. We also remark that knowing easy tasks to start training is not a strong assumption. Instead, it is very common in the curriculum learning literature.
>
> | | | | | | |
> |:- |:-:|:-:|:-:|:-:|:-:|
> |Simple-Spread (n=4)|VACL|GoalGAN|GoalGAN with easy cases|AMIGo|AMIGo with easy cases|
> |coverage rate % |$96.0 \pm 1.9 $|$51.0\pm31.8$|$74.4\pm15.3$|$0.27\pm0.07$|$0.27\pm0.06$|
> | | | | | | |
>
> 6. Clarity: “How were the continuous goals $\phi$ defined”
>
> $\phi$ is a vector that contains positions of agents and landmarks in Simple-Spread, and positions of agents, balls, and landmarks in Push-Ball. In Lock-and-Return, it contains positions of agents and boxes (without birthplaces). In Use-Ramp, we concatenate positions of the hider, boxes, and the ramp to get $\phi$. Regarding “easy cases”, we consider those cases where agents are near landmarks in Simple-Spread, and agents, balls, and landmarks are close to each other in Push-Ball, as easy tasks. As for Lock-and-Return, easy cases have agents near birthplaces, and boxes randomly placed near them. In Ramp-Use, easy cases have the ramp right next to the wall, and agents located next to the ramp. We will add these details to the final version.
>
> 7. "Choice of seed task" in Section 4.1.2 seems to have an error. ”
>
> Thank you so much for pointing this out. Yes, it is an error. The correct statement is that Exp.Act.w.eval performs better than Exp.Act. We will correct this.
>
> 8. “Related work”
>
> We will revise our paper to include these papers in the related work section.

---

> > ### Comment · Reviewer_spUJ · 2021-08-12
> > **Revised Review**
> >
> > After reading the author comments, I decided to improve my score to a weak accept, hoping that the authors incorporate the outlined clarifications and additional experiments. In particular, the paper would benefit from:
> >
> > * A hint to the experiment with a fixed number of agents for VACL in the appendix
> > * A rewriting of the rejection sampling method, clarifying how it is applied in the algorithm and clarifying the corresponding experiment that verifies its necessity in highly constrained task spaces. Also a mentioning of the oracle for rejection sampling could help readers.
> > * Inclusion of the results on combining GoalGAN with the initial knowledge of easy tasks (can be in the appendix with a reference to it in the main paper). Maybe also evaluate this combination in Push-Ball (n=2) for the sake of completeness (could then be added to Figure 10).
> > * The definition of the goals $\phi$ used in the experiments (can again be in the appendix and simply referenced).
> > * Inclusion of the additional references and a discussion of the differences w.r.t. the method presented therein
> >
> > Note that if I could ensure that all these changes made it into the paper, I would even go for a clear accept. However, this is unfortunately not possible.

---

> > > ### Author Response · Authors · 2021-08-12
> > > **Great Suggestions**
> > >
> > > We would like to express our appreciation again for all the great suggestions. They are really helpful.
> > >
> > > We promise to incorporate all these comments --- they will all make this paper a much better one!

---

> > > ### Author Response · Authors · 2021-08-30
> > > **Additional experiments**
> > >
> > > We provide additional experiments to evaluate GoalGAN and AMIGo with easy tasks in Push-Ball (n=2) in the following table and our algorithm still outperforms other methods with a clear margin.
> > >
> > > Moreover, we are working on our paper to incorporate all these comments and clarify the confusion.
> > >
> > > | | | | | | |
> > > |:----|:----:|:----:|:----:|:----:|:----:|
> > > |Push-Ball (n=2) |VACL|GoalGAN |GoalGAN with easy cases|AMIGo|AMIGo with easy cases|
> > > |coverage rate|$95.4\pm1.7\%$|$23.4\pm6.2%$|$14.8\pm5.0%$|$47.6\pm10.0%$|$56.7\pm10.5%$|
> > > |  |  |  |  | |

---

### Official Review · Reviewer_9Js4 · 2021-07-14

**Rating:** 6
**Confidence:** 3

**Summary:**

This paper proposed a variational automatic curriculum learning framework for solving goal-conditioned cooperative multiagent reinforcement learning problems. In this framework, the learning objective is decomposed into two parts: policy learning on current task distribution, and curriculum update to a new task distribution. The curricula are trained based on two components, including task configuration and the number of agents. The experimental results demonstrate the proposed method can achieve better performance than a few baselines with light weight computation resources.

**Ethical Concerns:**

I don't any ethical concerns.

**Limitations And Societal Impact:**

The author has provided some discussion regarding the limitation of their work. Additional limitations I see for this work are discussed in the Main Review.

**Main Review:**

In all, this paper studied an important method for solving cooperative multiple agent sequential decision-making tasks by using curricula learning. The authors tried to develop and analyze the solution method in a rigorous way. The authors also provide some good discussion on the connection to existing work. The method is tested over two environments and shown to outperform a few baselines. However, the motivation and some details related to the claimed contributions are not clearly explained, and the problems settings considered seems restrictive to homogenous agents.

Specifically, it is a little unclear to me why Stein variational gradient descent is necessarily better than regular variational inference method. It might be better to provide some more background information.

There are some technical details not clearly explained. For example, In section 2.2.1, “to update the particles, we consider an incremental transform T(\phi)=…”. Why it is necessary to introduce this transform? This seems to be one of the main ingredients for deriving the gradient update of task distribution. It might better if some design choices can be justified.

What is the number of particles used for performing Stein variational inference? How sensitivity is results to it?

Another limitation is that the paper only considered homogenous agents and learn a shared goal-conditioned policy. It might worth discussing how the approach can be generalized to heterogenous agents’ cases, since many realworld problems involve collaboration among different types of agents.
------ after rebuttal------
I think the author should explicit demonstrate that the method can work on problems with heterogenous agents. Given that the rebuttal has addressed most of my concerns, I would like to increase my score to 6.

**Time Spent Reviewing:**

5h

---

> ### Author Response · Authors · 2021-08-09
> **We will revise our paper to clarify the confusion**
>
> Thanks for your valuable comments. We will put more discussions into the updated paper for easier understanding.
>
> 1. “why Stein variational gradient descent is necessarily better than regular variational inference”
>
> Stein variational gradient descent is yet another widely adopted general-purpose variational inference framework. It is _unnecessarily_ better than classical mean-field variational inference or VAE-style framework. They both have pros and cons. Particularly in our use case of curriculum learning, Stein variational inference fits the best for two reasons. First, in the curriculum learning setting, the task space can be highly multi-modal, the use of particle approximation can be more flexible and represents the true underlying distribution better than a deep gaussian approximation as used in VAE. Second, whenever the policy is updated, the variational distribution needs to change accordingly. Classical variational inference framework requires re-training or fine-tuning of the proposal distribution $q(\phi)$, which can be particularly expensive when using a neural proposal distribution. By contrast, assuming a state-based object-centric task space available, a particle-based representation can efficiently adapt with a single SVGD step, i.e., perturb each particle towards the gradient direction, which is significantly more efficient. We remark that in our experiment section, we compare our particle-based framework with GoalGAN, which utilizes a neural task generator. GoalGAN can be considered as a method that uses a GAN to fit the proposal distribution. Empirically, GoalGAN requires substantially more samples to learn the task sampler. We will clarify this better in the paper.
>
> 2. “Why it is necessary to introduce the incremental transform T(\phi)=...”
>
> We introduce the incremental transform to rigorously formulate the process of curriculum learning so that we can apply Stein variational gradient descent here. Note that $q(\phi)$ denotes the distribution of training tasks in each curriculum learning iteration. As training proceeds, $q(\phi)$ adapts accordingly w.r.t. the policy. $T(\phi)$ formulates this adaptation process of $q(\phi)$ by using $\epsilon f(\phi)$ to denote the difference between the old and the adapted proposal distribution $q$ over the consecutive two curriculum learning iterations. In this formulation, when the policy is updated by a MARL algorithm, the proposal distribution $q(\phi)$ will be updated by $\phi\gets T(\phi)$ via SVGD.
>
> 3. “number of particles used for SGVD” and “sensitivity”
>
> We choose 4000 particles to approximate $q(\phi)$ in all our experiments. Note that with value quantization, we categorize $q(\phi)$ into an active set $Q_{act}$ and a solved set $Q_{sol}$ w.r.t. the task values. Each set uses 2000 particles. Regarding “sensitivity”, we additionally perform ablation studies with different numbers of particles in Simple-Spread with $n=4$ and the results in the following table show that we do need sufficient particles to approximate $q(\phi)$. Insufficient particle size (e.g., 400) results in inaccurate estimation and poor performance. The results are robust when particle size is at least 1000.
>
> | | | | | |
> |:----|:----:|:----:|:----:|:----:|
> |numer of particles|400|1000|4000|8000|
> |coverage rate %|$74.0\pm11.5$|$96.2\pm1.2$|$96.0\pm1.9$|$97.0\pm1.4$|
> |  |  |  |  |
>
> 4. “only consider homogeneous agents”
>
> VACL is a curriculum learning algorithm that only operatives over the task space. So it can be naturally combined with any MARL algorithm that works with heterogeneous agents. We primarily consider cooperative MARL tasks with policy sharing since most existing benchmarks adopt this homogeneous setting. Hence, we follow this literature convention. We will add another experiment domain with heterogeneous agents in our final version.

---

> > ### Comment · Reviewer_9Js4 · 2021-08-30
> > **Thanks**
> >
> > Thanks for the authors' rebuttal. I think the author should explicit demonstrate that the method can work on problems with heterogenous agents. Given that the rebuttal has addressed most of my concerns, I would like to increase my score to 6.

---

> > > ### Author Response · Authors · 2021-09-03
> > > **Additional Experiments**
> > >
> > > We additionally conduct experiments in Speaker-Listener which is one of the basic tasks in the MADDPG[1] paper. This task consists of two cooperative agents, a speaker, and a listener,  and three landmarks of differing colors. The speaker and listener obtain +1 reward when the listener covers the correct landmark. However, while the listener can observe the relative position and color of the landmarks, it does not know which landmark it must navigate to. Conversely, the speaker’s observation consists of the correct landmark color, and it can produce a communication output that is observed by the listener. Thus, the speaker must learn to output the landmark color based on the motions of the listener.
> > > In the following table, we train each algorithm for 30M timesteps and find that our algorithm can exactly work on the problem with heterogeneous agents.
> > >
> > > | | | | | | |
> > > |:----|:----:|:----:|:----:|:----:|:----:|
> > > |Speaker-Listener |VACL|Unif |RCG|GoalGAN|AMIGo|
> > > |coverage rate|$99.7\pm0.5\%$|$54.1\pm1.0%$|$99.6\pm0.4%$|$95.3\pm2.6%$|$55.1\pm0.9%$|
> > > |  |  |  |  |  |  |
> > >
> > > [1] Ryan Lowe, Yi Wu, et al. "Multi-Agent Actor-Critic for Mixed Cooperative-Competitive Environments." Neural Information Processing Systems, 2017.

---

> ### Author Response · Authors · 2021-08-29
> **A Gentle Reminder**
>
> Dear Reviewer 9Js4,
>
> Since the end of discussion period is approaching, we hope our explanations and additional results are sufficient to address your concerns. If there is anything we can provide for further clarification, please let us know and we are more than happy to make our contributions more clear.
>
> Best

---

### Official Review · Reviewer_TiSD · 2021-07-17

**Rating:** 7
**Confidence:** 5

**Summary:**

This paper introduces a curriculum learning method “VACL” to solve the multi-agents RL problems. The main idea of this paper is to train RL agents on an easy-to-hard task sequence based on an auto-generated curriculum.

**Limitations And Societal Impact:**

yes

**Main Review:**

The main idea of this paper is similar to some other curriculum RL methods, such as GoalGAN, setter and solver (https://arxiv.org/abs/1909.12892 ) and AMIGo. Due to the results of the experiments, this method achieves a great improvement on all previous work in a hide-and-seek environment. Meanwhile, VACL also achieves good performance on other tasks.

Originality: The main idea of this paper is similar to the GoalGAN and AMIGo. However, the curriculum generator is not as tricky as the mentioned methods to learn. The author uses a smooth function to represent the learning progress of a specific task instead, which achieves a higher sample efficiency. The essential part of this method is to use an incremental transform \T to simplify q(\phi). Due to the results, it’s obvious that this transform is available.

Clarity: I think this paper is written well. The expansion of the theoretical part is very detailed and logically fluent. The details of the experiments seem sufficient.

However, here are some questions and suggestions for this paper:

Questions:
1. In algorithm 1. \geteasy(n_{0}). How to define easy tasks for agents? Are these easy tasks picked by humans? What is the difference of two \geteasy used in \geteasy(n_0) and \geteasy(n_k+1)?
2. Section 3.4 is not sufficient enough. Considering that the main idea of VACL is to sample an easy-to-hard curriculum for agents to learn, and also the evaluation of the difficulty of tasks is also based on the V. Why not add the method mentioned in section 4.1 to section 3.4? It causes me so puzzled when I first read section 4.1.
3. Does the forgetting phenomenon occur in the experiments? If yes, how to solve it? I did not find any restrictions or guarantees if the agent forgets some tasks that have been resolved.
4. For empirical, I think V cannot fully express the value of learning on the current task. Maybe you can consider another metric to move tasks into Q_{act}, such as \delta V. Of course, this is just a possible optimization.
5. How to ensure the upper and lower bounds of "f" used in eq.4?

Overall, the paper offers a possible solution for evaluating the difficulty of tasks.

=== Post Rebuttal ===

The paper studies an interesting problem. Its claims are well supported from both theoretical and experimental perspectives.




**Time Spent Reviewing:**

4

---

> ### Author Response · Authors · 2021-08-09
> **Thanks for your comments**
>
> We appreciate your valuable comments.
>
> 1. Definition of “\GetEasy(n)”.
>
> The “\GetEasy(n)”  function takes in the parameter $n$, i.e., the number of agents (entities) in the environment (e.g., the initial agent size $n_0$ or $n_k$ after $k$ entity progression iterations), and produces samples of easy-to-learn tasks where the curriculum learning process starts with. This requires domain knowledge and is picked by humans. Such a minimal requirement is a very common assumption in curriculum learning literature. In our applications, easy tasks are defined by initial configurations where agents and objects are close to each other, which is a very common criterion for goal-conditioned tasks. Mode details are in Appendix D.1.
>
> 2. “Section 3.4”,
>
> We will revise our paper to add all the methods mentioned in section 4.1 to section 3.4.
>
> 3. “forgetting phenomenon”,
>
> Our framework will not induce the forgetting phenomenon. In our variational learning framework, the training tasks are sampled from the proposal distribution $q(\phi)$, which conceptually represents the *entire solvable task space*, i.e., tasks with sufficiently large values. Hence, the policy is forced to train with those tasks that are previously solved, which prevents the forgetting phenomenon.
> In our implementation, we use a set of task particles to approximate $q(\phi)$ and use value quantization for value approximation. We empirically notice that training on active tasks (Figure 1) leads to a faster learning process, so finally, we use 95% active tasks and 5% solved tasks to produce a training batch. Note that the technique of sampling a small ratio of previously solved tasks is often called past sampling in the existing literature as a common practice to prevent catastrophic forgetting. It is also worth mentioning that we did perform an ablation study that uses 100% active tasks for training in Simple-spread and Push-Ball, which makes no difference in performance. But in general, we believe that forgetting phenomena will occur without past sampling.
>
> 4. “V cannot fully express the value of learning”
>
> Thanks for the valuable suggestion. Here we use the value function to fully leverage the power of variational inference and we are still working on this towards more flexible evaluation metrics in our future work.
>
> 5. “the upper and lower bounds of ‘f’ ”
>
> The upper and lower bounds of $f^\star$ depend on the choice of the kernel function. In our application, we choose the RBF kernel (below Eq(5)), i.e., $k(\phi, \phi^{'}) = exp(-\frac{1}{h}||\phi-\phi^{'}||^2_2)$ so that $f^\star(\phi) =\Sigma_{\phi \in \mathcal{Q}_{act}} 2\frac{\phi-\phi'}{h}exp(-\frac{1}{h}||\phi-\phi^{'}||^2_2)$. Note that $0\le exp(-\frac{1}{h}||\phi-\phi^{'}||^2_2)\le 1$, so $f^\star$ is bounded by the diameter of the initial task configuration space. In our application, $\phi$ denotes the position of entities in the environment, which is bounded in every specific task.

---

> > ### Comment · Reviewer_TiSD · 2021-09-03
> > **Keep my original rating.**
> >
> > Thanks for the clarifications. After reading all reviewers' comments and the authors' responses, I decided to keep my original rating.

---

### Decision · Program_Chairs · 2021-09-27

**Decision:**

Accept (Poster)

**Comment:**

This work introduces a new curriculum learning algorithm for cooperative multi-agent reinforcement learning. All reviewers appreciated the method's novelty and that the paper was well-written. All found the theoretical motivation convincing and insightful. There were some initial concerns about the experimental results raised by reviewer spUJ, but they found the author's response convincing and thus increased their score.